# Essential role of submandibular lymph node dendritic cells in protective sublingual immunotherapy against murine allergy

Noriaki Miyanaga[1,2,8], Hideaki Takagi[1,3,8], Tomofumi Uto[1,3], Tomohiro Fukaya[1,3], Junta Nasu[1,4], Takehito Fukui[1,4], Yotaro Nishikawa[1,5], Tim Sparwasser[6], Narantsog Choijookhuu[7], Yoshitaka Hishikawa[7], Takeshi Nakamura[2], Tetsuya Tono[2] & Katsuaki Sato [1,3✉]

While sublingual immunotherapy (SLIT) is known as an allergen-specific treatment for type-1 allergies, how it controls allergic pathogenesis remains unclear. Here, we show the prerequisite role of conventional dendritic cells in submandibular lymph nodes (ManLNs) in the effectiveness of SLIT for the treatment of allergic disorders in mice. Deficiency of conventional dendritic cells or CD4$^+$Foxp3$^+$ regulatory T (T$_{reg}$) cells abrogates the protective effect of SLIT against allergic disorders. Furthermore, sublingual antigenic application primarily induces antigen-specific CD4$^+$Foxp3$^+$ T$_{reg}$ cells in draining ManLNs, in which it is severely impaired in the absence of cDCs. In ManLNs, migratory CD11b$^+$ cDCs are superior to other conventional dendritic cell subsets for the generation of antigen-specific CD4$^+$Foxp3$^+$ T$_{reg}$ cells, which is reflected by their dominancy in the tolerogenic features to favor this program. Thus, ManLNs are privileged sites in triggering mucosal tolerance mediating protect effect of SLIT on allergic disorders that requires a tolerogenesis of migratory CD11b$^+$ conventional dendritic cells.

[1] Division of Immunology, Department of Infectious Diseases, Faculty of Medicine, University of Miyazaki, 5200 Kihara, Kiyotake, Miyazaki 889-1692, Japan. [2] Department of Otolaryngology, Head and Neck Surgery, Faculty of Medicine, University of Miyazaki, 5200 Kihara, Kiyotake, Miyazaki 889-1692, Japan. [3] Japan Agency for Medical Research and Development (AMED), 1-7-1 Otemachi, Chiyoda-ku, Tokyo 100-0004, Japan. [4] Department of Oral and Maxillofacial Surgery, Faculty of Medicine, University of Miyazaki, 5200 Kihara, Kiyotake, Miyazaki 889-1692, Japan. [5] Department of Dermatology, Faculty of Medicine, University of Miyazaki, 5200 Kihara, Kiyotake, Miyazaki 889-1692, Japan. [6] Department of Medical Microbiology and Hygiene, University Medical Center, Johannes Gutenberg University, 55131 Mainz, Germany. [7] Division of Histochemistry and Cell Biology, Department of Anatomy, Faculty of Medicine, University of Miyazaki, Miyazaki 889-1692, Japan. [8]These authors contributed equally: Noriaki Miyanaga, Hideaki Takagi. ✉email: katsuaki_sato@med.miyazaki-u.ac.jp

Dendritic cells (DCs) are essential antigen-presenting cells (APCs) that play critical roles in orchestrating and regulating the immune system, linking innate and adaptive immunity[1–3]. DCs serve as sentinels in lymph nodes and peripheral tissues, recognizing the presence of invading pathogens through various pattern-recognition receptors to secrete multiple cytokines for the induction of inflammatory responses[1–3]. Subsequently, they process microbial antigens and present their antigenic peptides in the context of major histocompatibility complex (MHC) class I (MHC I) and MHC II in conjunction with costimulatory molecules and cytokines to naive T cells to initiate primary T-cell responses. DCs comprise heterogeneous subsets, functionally classified into conventional DCs (cDCs) and plasmacytoid DCs (pDCs)[1–3]. cDCs display a unique capacity to prime naive T cells to generate various types of effector T ($T_{eff}$) cells owing to the prominent expressions of MHC and costimulatory molecules[1–3]. In contrast, pDCs are specialized in the secretion of type-I interferon (IFN) upon recognition of viral nucleic acids through endosomal toll-like receptors 7/9, and this process contributes to the initiation of antiviral responses[3,4]. Conversely, DCs are critical for the maintenance of immune homeostasis by promoting immune tolerance via mechanisms including clonal deletion and anergy of antigen-specific T cells, as well as active immune suppression by $CD4^+Foxp3^+$ regulatory T ($T_{reg}$) cells that include self-reactive thymic-derived naturally occurring $T_{reg}$ ($nT_{reg}$) cells and antigen-specific peripherally induced $T_{reg}$ ($pT_{reg}$) cells generated from naive $CD4^+$ T cells[3].

Recent studies in the ontogeny of cDCs reveal two lineages of lymphoid-resident cDCs and non-lymphoid tissue-resident cDCs migrating to lymph nodes (migratory cDCs) that differ in their distinct developmental pathway and function[1–3]. Mouse lymphoid-resident cDC subsets in spleen are divided on the basis of the expression of CD8α, whereas migratory cDCs are defined by mutually exclusive surface expression of the integrins CD103 ($α_E$) and CD11b ($α_M$), although cDC subsets in the small intestinal lamina propria (siLP) express both of these markers[1–3]. The first lineage, referred to as cDC1, encompasses $CD8α^+$ cDCs and $CD103^+$ cDCs, and their development requires several transcription factors, including basic leucine zipper transcription factor ATF-like 3 (Batf3) and IFN regulatory factor (IRF)8[1–3]. The second lineage of cDCs (cDC2) consists of $CD8α^-$ cDCs and $CD11b^+$ DCs, and they develop dependently of IRF4[1–3]. cDC1 are specialized in the cross-presentation of antigens for cytotoxic T lymphocytes-responses and driving T-helper ($T_H$)1 cell responses to intracellular pathogens[1–3]. In addition, cDC2 have a non-redundant role in promoting $T_H$2 cell- and $T_H$17 cell responses to extracellular pathogens[1–3].

The mucosal surface in whole body is continuously exposed to enormous variety of foreign materials. In the gastrointestinal tract, the mucosal immune system maintains a balance between protective responses against potentially pathogenic microbes and unresponsiveness to harmless dietary constituents and commensal flora, a phenomenon known as oral tolerance[5]. Different from cDC subsets in other tissues, $CD103^+$ cDCs in the siLP are subdivided into three ontogenetically distinct subsets based on the expression of CD103 and CD11b, $CD103^+CD11b^-$ cDC1, $CD103^+CD11b^-$ cDC2 and $CD103^-CD11b^+$ cDC2[6,7]. Upon sampling luminal antigens, LP $CD103^+$ cDCs migrate to mesenteric lymph nodes (MesLNs) where are the key sites in initiating oral tolerance[7,8]. These mucosal $CD103^+$ cDCs are endowed with tolerogenic properties, including the production of large quantities of retinoic acid (RA) and transforming growth factor (TGF)-β to promote the intestinal emergence of $CD4^+Foxp3^+$ $pT_{reg}$ cells to establish oral tolerance[7–11].

Allergen-specific immunotherapy is an allergen-specific treatment for type-1 allergies with long-lasting effects through the reorientation of inappropriate immune responses in allergic patients[12,13]. Subcutaneous immunotherapy (SCIT) is conventional way of allergen-specific immunotherapy with subcutaneous administration of the specific causative allergen in an incremental dose regimen[12,13]. The proposed mechanisms of the action of SCIT include the redirection of allergen-specific $T_H$2-responses to $T_H$1-responses, leading to the suppression of allergen-specific immunoglobulin (Ig)E and induction of $IgG_4$ from B cells as well as the inhibition of mast cells, basophils, and eosinophils[12,13]. However, SCIT can be associated with severe side effects, including anaphylactic shocks since it requires subcutaneous route of multiple injections of allergen[14].

Sublingual immunotherapy (SLIT) is established in IgE-dependent respiratory allergies, such as allergic rhinitis and rhinoconjuctivitis, in clinical setting to grass or tree pollens, as well as house dust mites as an effective and safety profile with rare incidence of anaphylaxis convenient alternative strategy to SCIT, whereas the clinical efficacy of its application for other allergic disorders remains unclear[15,16]. SLIT is implicated to operate by acting on the sublingual mucosa and inhibit allergic immune responses, and the mechanisms possibly involve the immunogenic shift from $T_H$2- to $T_H$1-response and/or the tolerogenic profile associated with $CD4^+Foxp3^+$ $T_{reg}$ cells[17–22]. Given the postulated mechanisms, how SLIT controls allergic response is less well understood than those of SCIT. Although several subsets of macrophages and cDCs in sublingual mucosal tissues have been considered to capture antigens to generate $T_H$1 cells and/or $CD4^+Foxp3^+$ $T_{reg}$ cells following sublingual antigenic application[23–27], how migratory cDCs in draining submandibular lymph nodes (ManLNs) control allergic immune responses currently remains unclear even in the mouse model of SLIT.

In this study, we demonstrate the impact of the tolerogenesis of migratory ManLN cDCs on the effectiveness of SLIT for the abortive allergic immune responses leading the protection against the development of acute and chronic allergic disorders.

## Results

**cDCs are required for the protective effect of SLIT on $T_H$2-driven allergic pathogenesis.** To clarify the contribution of cDCs to the effectiveness of SLIT for protecting against $T_H$2-mediated allergic airway responses, we use CD11c-diphtheria toxin (DT) receptor (DTR)/enhanced green fluorescent protein (EGFP) transgenic (Tg) mice[28,29], in which $CD11c^{high}SiglecH^-$ cDCs, but not $CD11c^{int}SiglecH^+$ pDCs, were depleted after DT injection (referred to as cDC-ablated mice) (Supplementary Fig. 1a–c). Intranasal administration with ovalbumin (OVA) protein after intraperitoneal (i.p.) immunization with OVA protein plus alum adjuvant elicited typical asthmatic symptoms with accumulation of immune cells in bronchoalveolar lavage fluid, serum production of OVA-specific $IgG_1$ and IgE, augmentation of airway hyperresponsiveness to methacholine as well as infiltration of inflammatory cells and mucus production in the lung in wild-type (WT) mice (Fig. 1a, b, e–l and Supplementary Fig. 1d). On the other hand, the sublingual administration with OVA protein before immunization ameliorated these $T_H$2-deriven responses and airway allergic pathogenesis in WT mice (Fig. 1a, b, e–l). However, the elimination of cDCs abrogated the protective effect of SLIT on $T_H$2-mediated airway inflammation (Fig. 1c, d and Supplementary Fig. 2).

The application of SLIT has been reportedly limited in the treatment of allergic rhinitis and rhinoconjuctivitis in humans[15,16]. To clarify whether SLIT is effective for the treatment of other $T_H$2-mediated allergic disorders, we investigated the protective effect of SLIT on the development of food allergy and systemic anaphylaxis. The consecutive intragastoric (i.g.)

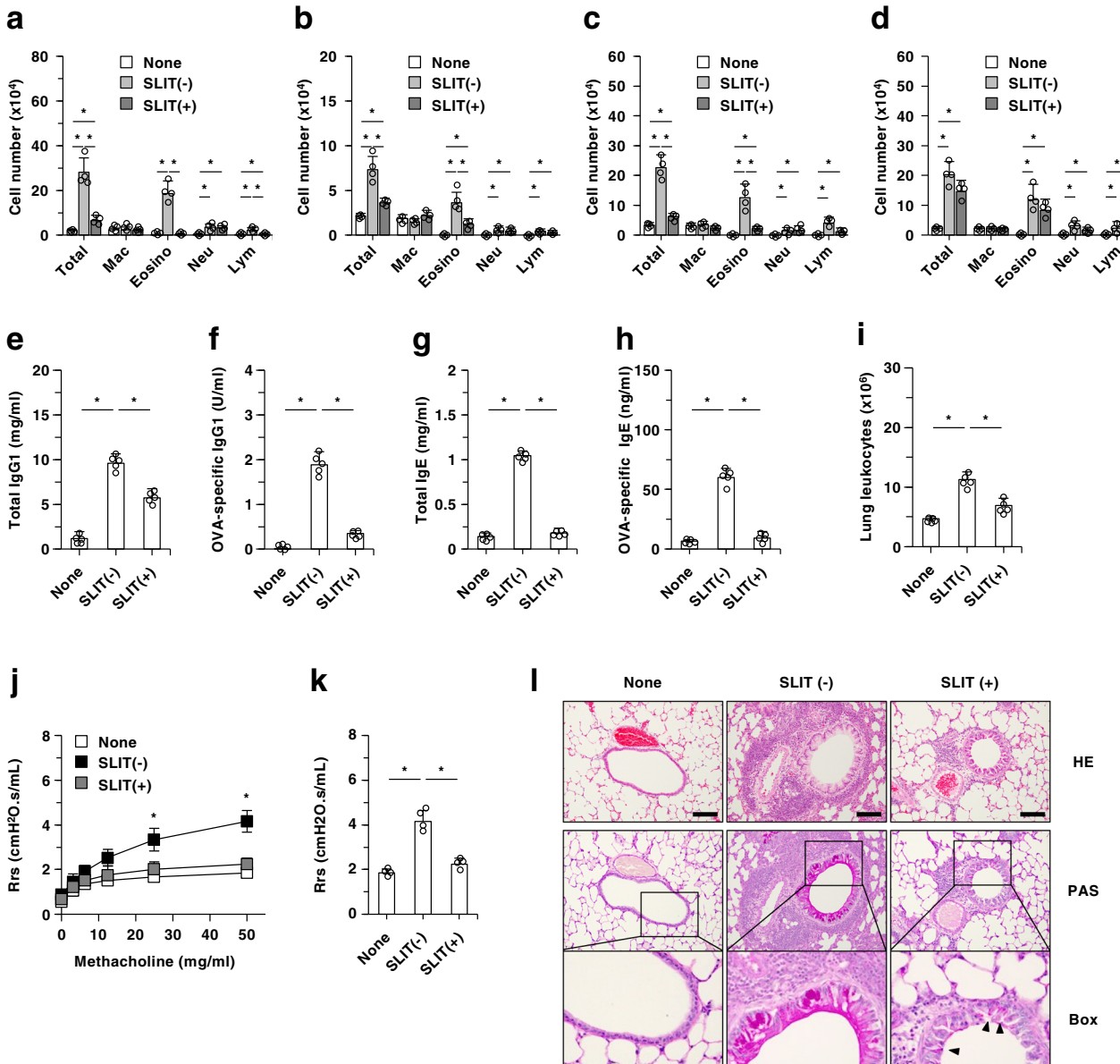

**Fig. 1 Deficiency of cDCs dampens the protective effect of SLIT on $T_H2$-mediated allergic airway inflammation.** WT mice and CD11c-DTR/EGFP mice that had been treated with PBS or DT were sublingually administered with PBS or OVA protein, and then systemically immunized with or without OVA protein at 7 and 14 days after SLIT. Subsequently, mice were intranasally sensitized with or without OVA protein at 10, 11, and 12 days after the last immunization, and BALF, serum and lung tissue were obtained at 13 days after the last sensitization. **a-d** Absolute cell numbers of BALF cells obtained from Balb/c WT mice **a**, B6 WT mice **b**, and CD11c-DTR/EGFP mice that had been treated with PBS **c** or DT **d**. *Mac*, macrophages; *Eosino*, eosinophils; *Neu*, neutrophils; *Lym*, lymphocyte. **e-h** Serum production of total $IgG_1$ **e**, OVA-specific-$IgG_1$ **f**, total IgE **g**, and OVA-specific IgE **h** in Balb/c WT mice. **i** Absolute cell numbers of lung leukocyte obtained from Balb/c WT mice. **j, k** Respiratory airway function was analyzed for changes in resistance of the respiratory system (Rrs) in response to increasing doses (**j** 3.125–50 mg/ml) or the indicated dose (**k** 50 mg/ml) of inhaled methacholine in Balb/c WT mice. **l** Hematoxylin and eosin (HE) and periodic acid–Schiff (PAS) sections obtained from the lung in Balb/c WT mice at low-magnification (×20) and higher magnification (×40). Bars indicate 100 μm. Arrow indicates mucus deposition. Data are obtained from five individual samples in a single experiment. *$P < 0.05$ compared with normal mice (none) or among groups. All data are representative of at least three independent experiments.

administrations with OVA protein after i.p. immunization with OVA protein plus alum adjuvant evoked food allergy, as indicated by diarrhea occurrence and fecal scoring as well as the massive infiltration of mast cells in small intestine (SI), whereas SLIT attenuated the development of these pathogenesis (Supplementary Fig. 3a–c). Although i.p. sensitization with OVA protein after systemic immunization with OVA protein plus alum adjuvant caused systemic anaphylaxis marked in the decline in body temperature and histamine release in serum, as well as the production of OVA-specific $IgG_1$ and the binding of basophils

with OVA protein, SLIT ameliorated these anaphylactic symptoms (Supplementary Fig. 4a, b, d–j). However, the deficiency of cDCs abrogated the protective effect of SLIT on the development of systemic anaphylaxis (Supplementary Fig. 4c).

Collectively, these results indicate that SLIT is effective for the protection against the initiation and progression of allergic airway inflammation as well as food allergy and systemic anaphylaxis, whereas the absence of cDCs impairs protective effect of SLIT on the development of these $T_H2$-mediated allergic pathogenesis.

**SLIT suppresses antigen-specific $T_H2$-mediated immune responses**. To clarify the mechanism how SLIT controls the immune responses for the protection against $T_H2$-mediated allergic pathogenesis, we assessed the generation of the pathogenic $T_H2$ cells expressing ST2 known as the component of interleukin (IL-33) receptor[30] in the lung. The asthmatic mice displayed higher frequency and absolute cell number of $ST2^+$ $CD4^+$ T cells in the lung than normal mice, while SLIT markedly diminished the generation of $ST2^+CD4^+$ T cells in asthmatic mice (Fig. 2a–c). Similar results were observed in the suppressive effect of SLIT on the intestinal generation of $ST2^+CD4^+$ T cells in food allergic mice (Supplementary Fig. 3d–f). Furthermore, $CD4^+$ T cells exhibited higher productions of IL-4, IL-5, and IL-13, but not IFN-γ, in the lung of asthmatic mice than normal mice, whereas SLIT suppressed their cytokine productions in asthmatic mice (Fig. 2d–h). Similarly, $CD4^+$ T cells displayed prominent productions of IL-4, IL-5, IL-13, and IFN-γ in draining lymph nodes of asthmatic mice when compared with normal mice, whereas SLIT reduced their cytokines productions in asthmatic mice (Supplementary Fig. 5a–e).

We also addressed the impact of SLIT on the formation of germinal center in secondary lymphoid tissues, allowing B-cell clonal expansion, Ig class switching, and affinity maturation for the generation of plasma cells and memory B cells[31]. As B cells within germinal centers are detected by peanut agglutinin (PNA) and GL7[31], fluorescent microscopic analysis of the spleen revealed that allergic mice exhibited the enhanced generation of $B220^+PNA^+$ germinal center B cells, whereas SLIT markedly decreased the emergence of germinal center B cells in allergic mice (Fig. 2i). Furthermore, allergic mice displayed a higher frequency of $PNA^+GL7^+$ cells and $IgG1^+$ cells within antigen-specific B cells than normal mice, whereas SLIT suppressed the generation of these cell populations in allergic mice (Fig. 2j–m).

We further examined the generation of follicular helper T ($T_{FH}$) cells because they have reportedly been involved in the formation of the germinal center[31]. Although allergic mice displayed the generation of $CXCR5^+PD-1^+$ $T_{FH}$ cells in spleen, SLIT markedly diminished their accumulation in asthmatic mice (Supplementary Fig. 5f, g).

Taken together, these results indicate that SLIT is effective for the inhibition of antigen-specific $T_H2$-cell responses and B-cell responses in allergic mice.

**Oral migratory cDCs transport sublingual antigens to draining ManLNs for $CD4^+$ T-cell priming**. To clarify the role of cDCs in the initiation of antigen-specific $CD4^+$ T-cell response following sublingual application of antigen, we adaptively transferred eFluor670-labeled $OT-II^+CD4^+$ T cells expressing the OVA-specific T-cell receptor (TCR)[4] into mice, sublingually administrated OVA protein, and monitored their antigen-specific division in various lymph nodes. Sublingual administration of OVA protein induced antigen-specific division of $OT-II^+CD4^+$ T cells in ManLNs known as draining lymph nodes of sublingual mucosa, but not in non-draining auricular lymph nodes (AurLNs) and MesLNs (Fig. 3a). In contrast, the ablation of cDCs markedly diminished antigen-specific division of $OT-II^+CD4^+$ T cells in ManLNs after sublingual application of OVA protein (Fig. 3b, c).

Because the transfer of fed antigens from $CX_3CR_1^+$ macrophages to $CD103^+$ cDCs has been reported in the intestinal mucosa for the establishment of oral tolerance[32], we examined the influence of the depletion of oral macrophages on the activation of antigen-specific $CD4^+$ T cells in ManLNs following sublingual antigenic application. Whereas the treatment with clophosome-A (CL-A) efficiently depleted oral macrophages, but

not cDCs, (Supplementary Fig. 6a–c), it had little effect on antigen-specific division of $OT-II^+CD4^+$ T cells after sublingual application of OVA protein (Supplementary Fig. 6d, e).

We further addressed the subsets of APCs that capture and present OVA protein in ManLNs following sublingual application of Alexa647-labeled OVA protein. In ManLNs, migratory MHC II (I-A/I-E)$^{hi}CD11c^{med}CD11b^+CD103^-$ cDCs (migratory $CD11b^+$ cDCs) retained OVA protein, whereas little or no retention was observed in other APCs, including migratory I-A/I-$E^{hi}CD11c^{med}CD11b^-CD103^+$ cDCs (migratory $CD103^+$ cDCs) and resident I-A/I-$E^{med}CD11c^{hi}$ cDCs (resident cDCs) (Fig. 3d and Supplementary Fig. 6f, g). On the other hand, the ablation of cDCs dramatically inhibited the accumulation of OVA protein in ManLNs after sublingual antigenic application (Fig. 3e, f).

Collectively, these results indicate that sublingual antigen is mainly retained in I-A/I-$E^{hi}CD11c^{med}CD11b^+$ cDCs to induce the proliferation of antigen-specific $CD4^+$ T cells in ManLNs.

**cDCs mediates de novo generation of antigen-specific $CD4^+$ $Foxp3^+$ $pT_{reg}$ cells in ManLNs upon sublingual antigenic application**. Although several immune deviations have been proposed to mediate the protective effect of SLIT on the development of allergic disorders in humans and rodents[17–22], how SLIT operates $T_H2$-deriven allergic pathologies remains unclear. We therefore examined the role of cDCs in the emergence of $CD4^+Foxp3^+$ $pT_{reg}$ cells in spleen, MesLNs, and ManLNs at 5 days after sublingual antigenic application. Sublingual administration of OVA protein induced the generation of antigen-specific $OT-II^+CD4^+Foxp3^{EGFP+}$ $pT_{reg}$ cells in ManLNs, but not spleen and MesLNs, when WT mice were adaptively transferred with $OT-II^+CD4^+Foxp3^{EGFP-}$ T cells[4] (Fig. 4a, b and Supplementary Fig. 7a). In contrast, the elimination of cDCs abrogated the generation of $OT-II^+CD4^+Foxp3^{EGFP+}$ $pT_{reg}$ cells after sublingual antigenic application (Fig. 4c, d). At 7 days after sublingual antigenic application, the accumulation of $OT-II^+CD4^+$ $Foxp3^{EGFP+}$ $pT_{reg}$ cells in spleen was observed, while their frequency was further enhanced in ManLNs (Fig. 4e, f). On the other hand, ManLN cDCs obtained from WT mice that had received sublingual administration of OVA protein, but not normal mice, generated $CD4^+Foxp3^{EGFP+}$ $T_{reg}$ cells from $KJ1–26^+CD4^+Foxp3^{EGFP-}$ T cells expressing the OVA-specific TCR (KJ1–26 clonotype)[8] (Supplementary Fig. 7b, c).

Taken together, these results indicate that sublingual antigenic application induces the generation of antigen-specific $CD4^+Foxp3^+$ $pT_{reg}$ cells in ManLNs, whereas the absence of cDCs abrogates their generation.

To assess whether $CD4^+Foxp3^+$ $pT_{reg}$ cells generated in ManLNs migrate to peripheral tissues, we examined the influence of FTY720, a potent inhibitor of lysophospholipid sphingoshine-1-phosphate (S1P) receptors (S1PRs) for leukocyte egress from lymph nodes[29], on their emergence in ManLNs and spleen. Upon sublingual antigenic application, the accumulation of $OT-II^+CD4^+Foxp3^{EGFP+}$ $pT_{reg}$ cells in spleen was attenuated by FTY720 treatment, whereas their retention was further increased in ManLNs (Fig. 4g, h), indicating that $CD4^+Foxp3^+$ $pT_{reg}$ cells generated in ManLNs mobilize into the peripheral tissues depending on S1PR-mediated signaling.

We further addressed whether antigen-specific $CD4^+Foxp3^+$ $T_{reg}$ cells is sufficient for the suppression of $T_H2$-mediated allergic pathogenesis. The adaptive transfer with $CD4^+CD25^+$ $T_{reg}$ cells, but not $CD4^+CD25^-$ T cells obtained from mice received sublingual antigenic application and $CD4^+CD25^+$ $T_{reg}$ cells from naive mice, inhibited the development of systemic anaphylaxis (Fig. 4i and Supplementary Fig. 7d). We also observed that $CD4^+CD25^+$ $T_{reg}$ cells, but not $CD4^+CD25^-$ T cells obtained

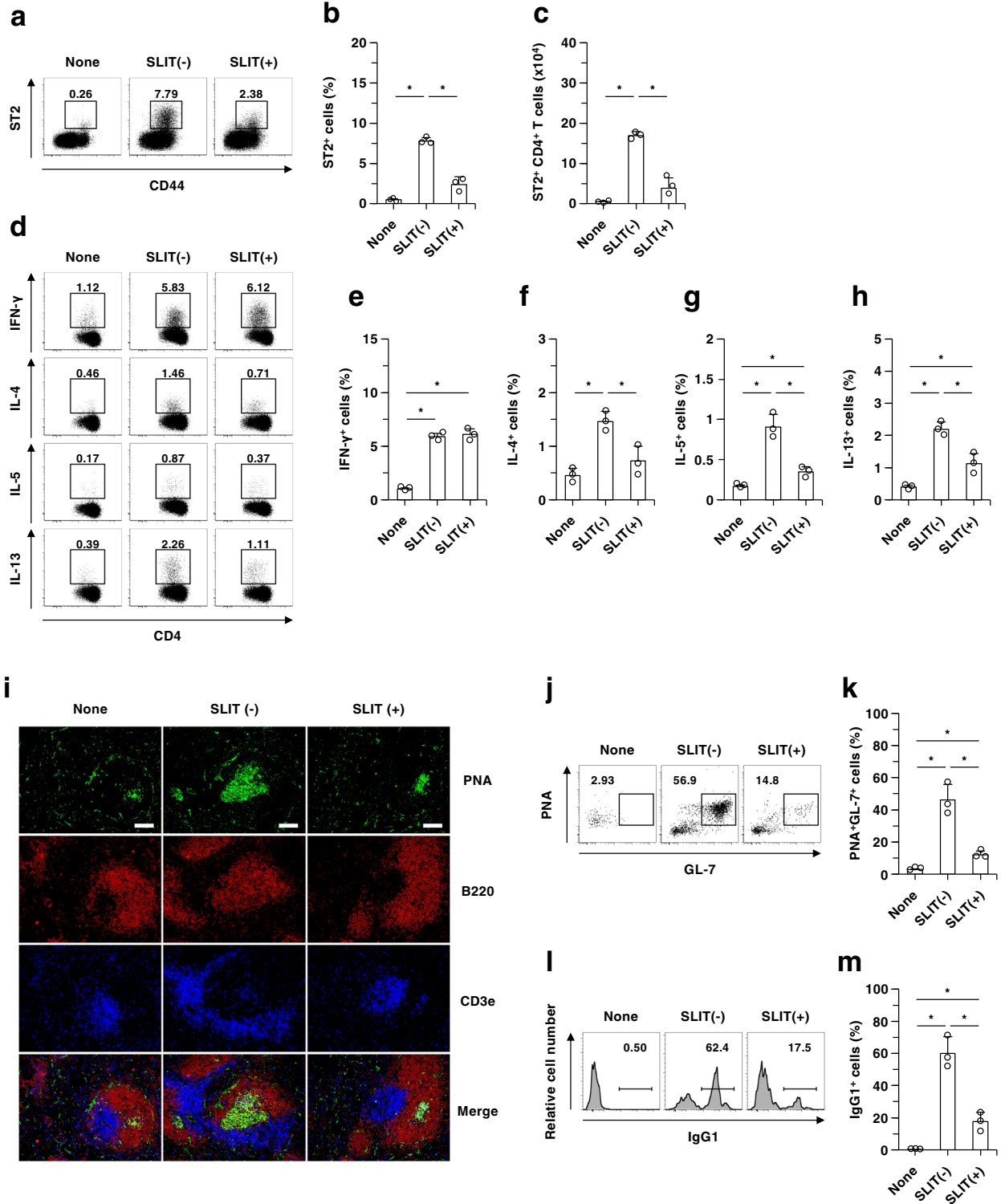

**Fig. 2 SLIT prevents T$_H$2-mediated allergic immune responses in lung of asthmatic mice.** WT mice were immunized as described in Fig. 1 for the induction of allergic airway inflammation. **a–c** Cell surface expression profile **a** and proportion **b** of ST2+CD44+ cells among CD4+ T cells, and absolute cell number **c** of CD4+ST2+CD44+ T cells in lung. **d–h** Intracellular cytokine expression profile **d** and proportion **e–h** of lung CD4+ T cells. **i** Frozen sections obtained from spleen were stained for PNA (green), B220 (red), and CD3ε (blue) at low-magnification (×20). Bars indicate 100 μm. **j–m** Cell surface expression profile **j**, **l** and proportion **k**, **m** of PNA+GL7+ **j**, **k**, and IgG$_1$+ **l**, **m** cells within OVA-specific B cells in the spleen. Data are obtained from five individual samples in a single experiment. *$P < 0.05$ compared with normal mice (none) or among groups. All data are representative of at least 3 independent experiments.

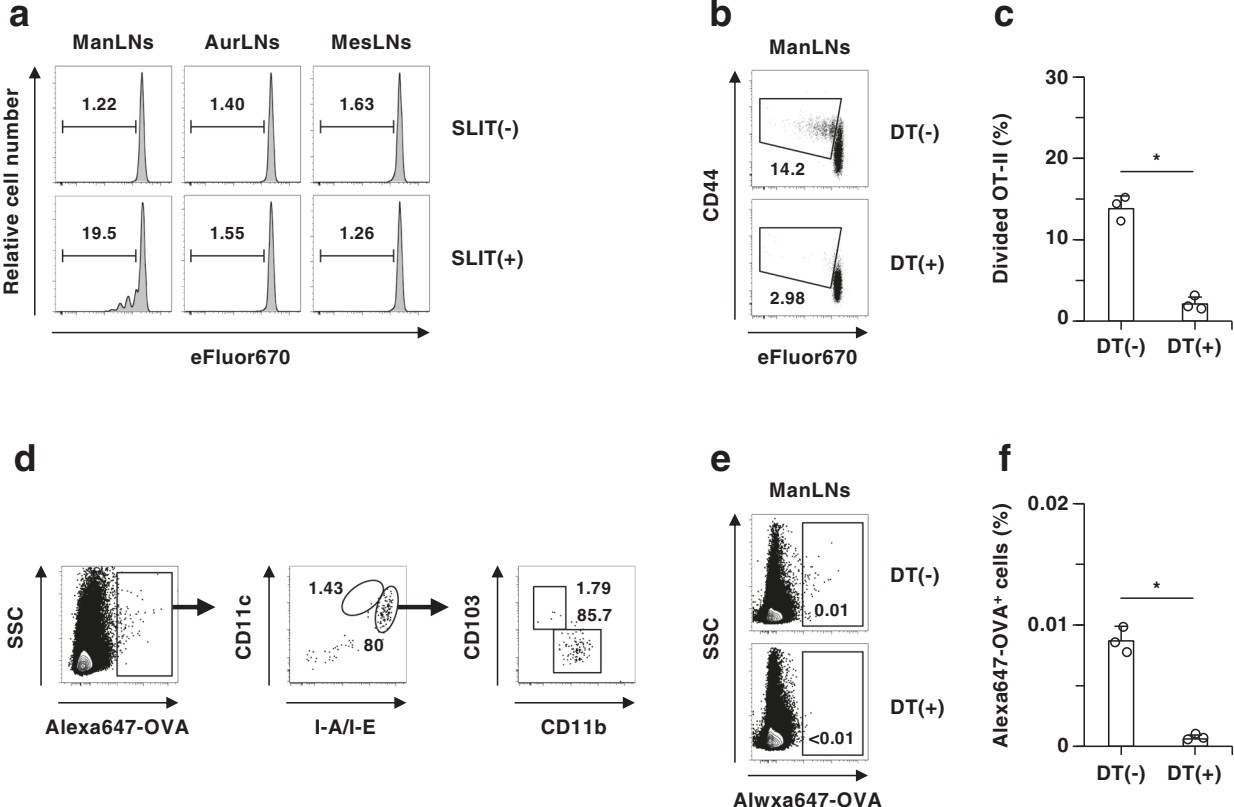

**Fig. 3 Deficiency of cDCs attenuates antigen-specific priming of CD4$^+$ T cells in the draining ManLNs. a–c** WT mice **a** and CD11c-DTR/EGFP mice that had been treated with PBS **b**, **c** or DT **b**, **c** were adoptively transferred with eFluor™ 670-labeled CD45.1$^+$OT-II CD4$^+$ T cells, and then mice were sublingually administered with OVA protein. Cell dividing profile **a**, **b** and proportion **c** of CD45.1$^+$OT-II CD4$^+$ T cells in ManLNs **a–c**, AurLNs **a**, and MesLNs **a** at 3 days after the administration. **d–f** WT mice **d** and CD11c-DTR/EGFP mice that had been treated with PBS **e**, **f** or DT **e**, **f** were sublingually administrated with AlexaFluor™ 647-labeled OVA protein. Cell surface expression profile **d**, **e** and proportion **f** of AlexaFluor™ 647-labeled OVA protein-binding cells among leukocytes in ManLNs at 18 h after the administration. Data are obtained from five individual samples in a single experiment. *$P < 0.05$ compared with CD11c-DTR/EGFP mice that had been treated with PBS indicated as DT(−). All data are representative of at least three independent experiments.

from mice received sublingual antigenic application exhibited a more inhibitory effect on antigen-specific proliferation of KJ1–26$^+$CD4$^+$ST2$^+$CD44$^+$ T cells than CD4$^+$CD25$^+$ T$_{reg}$ cells from naive mice (Supplementary Fig. 7e–g). On the other hand, the ablation of CD4$^+$Foxp3$^+$ T$_{reg}$ cells impaired the protective effect of SLIT on the development of systemic anaphylaxis in DEREG (depletion of T$_{reg}$) mice[33] following DT injection (Fig. 4J and Supplementary Fig. 7h, i).

Collectively, these results indicate that antigen-specific CD4$^+$Foxp3$^+$ T$_{reg}$ cells mainly mediate the protective effect of SLIT on the development of allergic disorders.

**ManLN migratory CD11b$^+$ cDCs contribute to de novo generation of antigen-specific CD4$^+$Foxp3$^+$ pT$_{reg}$ cells.** Although it has been shown that intestinal CD103$^+$ cDCs play an important in the generation of antigen-specific pT$_{reg}$ cells during the establishment of oral tolerance[7–11], the potential of cDC subsets in ManLNs to induce CD4$^+$Foxp3$^+$ pT$_{reg}$ cells remains unclear. We therefore compared the ability of cDC subsets in ManLNs as well as spleen and MesLNs to induce the differentiation of CD4$^+$Foxp3$^-$ T cells into CD4$^+$Foxp3$^+$ pT$_{reg}$ cells. Different from the superior capacity of CD103$^+$ cDCs to CD103$^-$ cDCs in the generation of CD4$^+$Foxp3$^+$ pT$_{reg}$ cells in intestinal tissues and other peripheral lymph nodes (PLNs)[3,7,10,11], migratory CD11b$^+$ cDCs exhibited a higher generation of antigen-specific

CD4$^+$Foxp3$^{EGFP+}$ T$_{reg}$ cells from CD4$^+$Foxp3$^{EGFP-}$ T cells than migratory CD103$^+$ cDCs and resident cDCs in ManLNs as well as splenic cDCs, and that was comparable to MesLN cDCs (Fig. 5a, b).

We also examined the characteristic features of ManLN cDCs for the generation of CD4$^+$Foxp3$^+$ pT$_{reg}$ cells. Each subset of migratory and resident cDCs exhibited the differential expression levels of the costimulary molecules, XCR1, Sirpα, IRF4, and IRF8 between ManLNs and MesLNs (Fig. 5c and Supplementary Fig. 8a, b). In ManLNs, migratory CD11b$^+$ cDCs exhibited a typical phenotype of cDC2 with the higher expressions of Sirpα and IRF4 and lower expressions of XCR1 and IRF8 than migratory CD103$^+$ cDCs and resident cDCs (Fig. 5c). Migratory CD11b$^+$ cDCs expressed higher levels of B7-H1 and B7-DC than migratory CD103$^+$ cDCs and resident cDCs (Fig. 5c), which were different from those in MesLN cDC subsets (Supplementary Fig. 8b). Furthermore, anti-B7-DC monoclonal antibody (mAb) partially abrogated their ability to generate KJ1–26$^+$CD4$^+$Foxp3$^{EGFP+}$ T$_{reg}$ cells (Supplementary Fig. 8c). Similar results were observed in the inhibitory effect of the blockade of B7-DC on the ability of MesLN cDCs to generate KJ1–26$^+$CD4$^+$Foxp3$^{EGFP+}$ T$_{reg}$ cells (Supplementary Fig. 8d). On the other hand, migratory CD11b$^+$ cDCs and migratory CD103$^+$ cDCs displayed a greater transcriptional expression of *Itgb8*, which is reportedly involved in the activation of TGF-β[7,9–11], than resident cDCs,

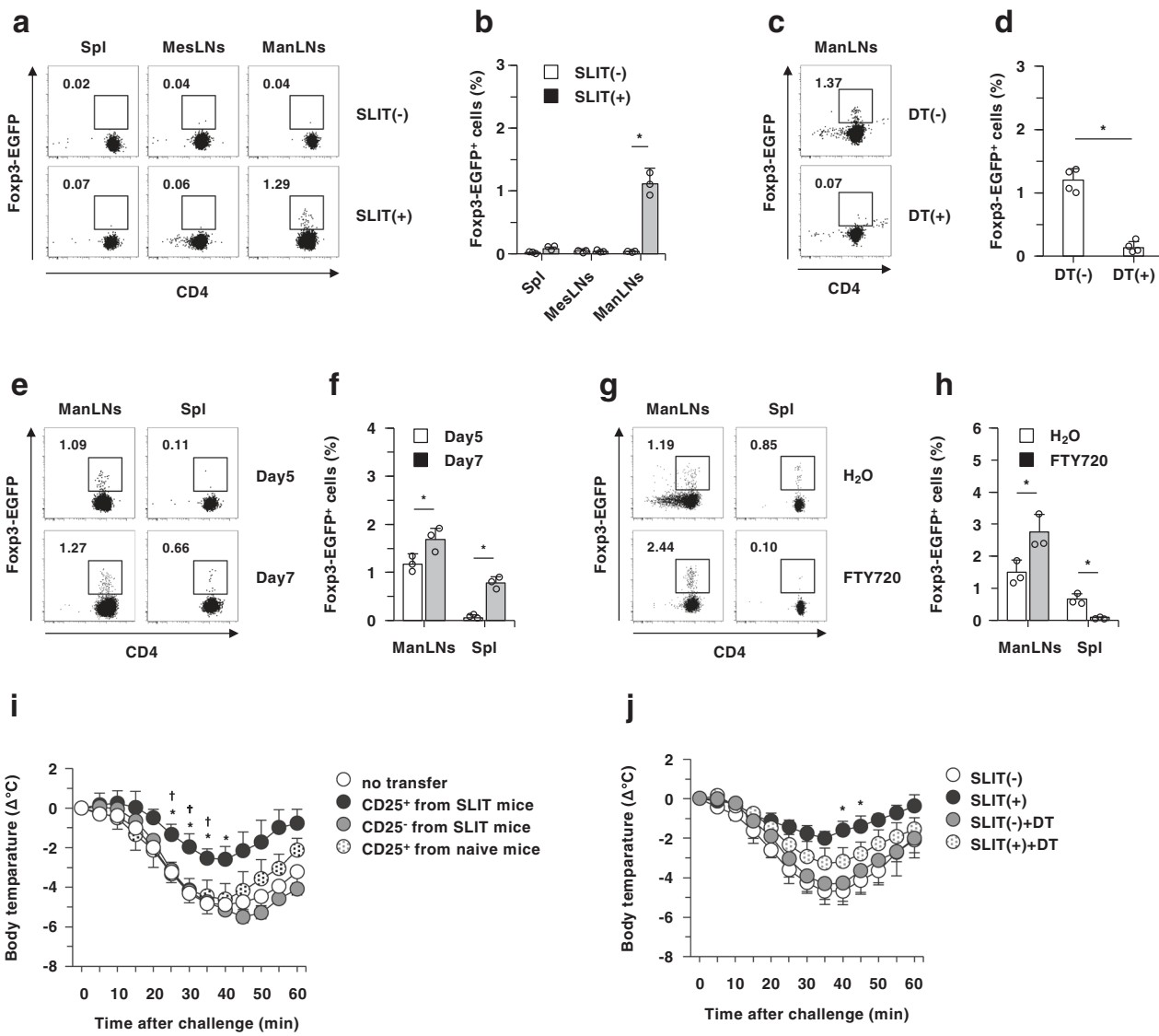

**Fig. 4 Deficiency of cDCs abrogates de novo generation of antigen-specific CD4$^+$Foxp3$^+$ pT$_{reg}$ cells in vivo. a–h** WT mice **a**, **b**, **e–h**, CD11c-DTR/EGFP mice that had been treated with PBS **c**, **d** or DT **c**, **d** were adoptively transferred with CD45.1$^+$OT-II CD4$^+$Foxp3$^{EGFP−}$ T cells, and then mice were sublingually administered with OVA protein. The expression of Foxp3$^{EGFP}$ on CD45.1$^+$OT-II CD4$^+$ T cells in spleen, MesLNs, and ManLNs were analyzed at 5 days **a–f** or 7 days **e**, **f** after the administration. **g**, **h** WT mice were daily administrated with or without FTY720, and the expression of Foxp3$^{EGFP}$ on CD45.1$^+$OT-II CD4$^+$ T cells in spleen and ManLNs were analyzed at 7 days after the administration. Cell surface expression profile **a**, **c**, **e**, **g** and proportion **b**, **d**, **f**, **h** of CD45.1$^+$OT-II CD4$^+$Foxp3$^{EGFP+}$ T cells. **i** CD4$^+$CD25$^+$ T cells or CD4$^+$CD25$^−$ T cells obtained from WT mice that had been treated with or without sublingual immunization of OVA were adoptively transferred into WT mice, and then mice were systemically immunized with OVA protein at 1 day and 8 days after the transfer. Subsequently, mice were challenged i.p. with OVA protein at 10 days after the last immunization, and the rectal temperature were monitored. **j** DEREG mice that had been treated with or without sublingual immunization of OVA were administrated with or without DT at 7 days after SLIT, and then mice were systemically immunized with OVA protein at 15 day and 22 days after DT injection. Subsequently, mice were challenged i.p. with OVA protein at 10 days after the last immunization, and the rectal temperature were monitored. Data are obtained from five individual samples in a single experiment. *$P < 0.05$ compared with normal mice **b** indicated as SLIT(−), CD11c-DTR/EGFP mice that had been treated with PBS indicated as DT (−) **d**, among groups **f**, **h** WT mice (no transfer) **i**, or DEREG mice immunized with OVA protein indicated as SLIT(−) **j**. †$P < 0.05$ compared with CD4$^+$CD25$^+$ T-cell transfer from naive mice **i**. All data are representative of at least three independent experiments.

whereas similar expression level of transcript of *Itgav* was observed in these DC subsets (Fig. 5d, e). Furthermore, migratory CD11b$^+$ cDCs not only displayed a marked expression of *aldh1a2*, which encode RA-generating enzyme retinal dehydrogenase 2 (RALDH2)[5], but also showed a prominent activity of aldehyde dehydrogenase (ALDH) compared with migratory CD103$^+$ cDCs and resident cDCs (Fig. 5f–i). In contrast, migratory CD103$^+$ cDCs exhibited a higher activity of ALDH than other DC subsets in MesLNs (Supplementary Fig. 8e, f).

Collectively, these results indicate that migratory CD11b$^+$ cDCs are superior to other cDC subsets for the generation of CD4$^+$Foxp3$^+$ pT$_{reg}$ cells in ManLNs that are different from intestinal tissues and other PLNs.

## Discussion
In this study, our findings propose that ManLNs are the primary sites in initiating mucosal tolerance that mediate the protective

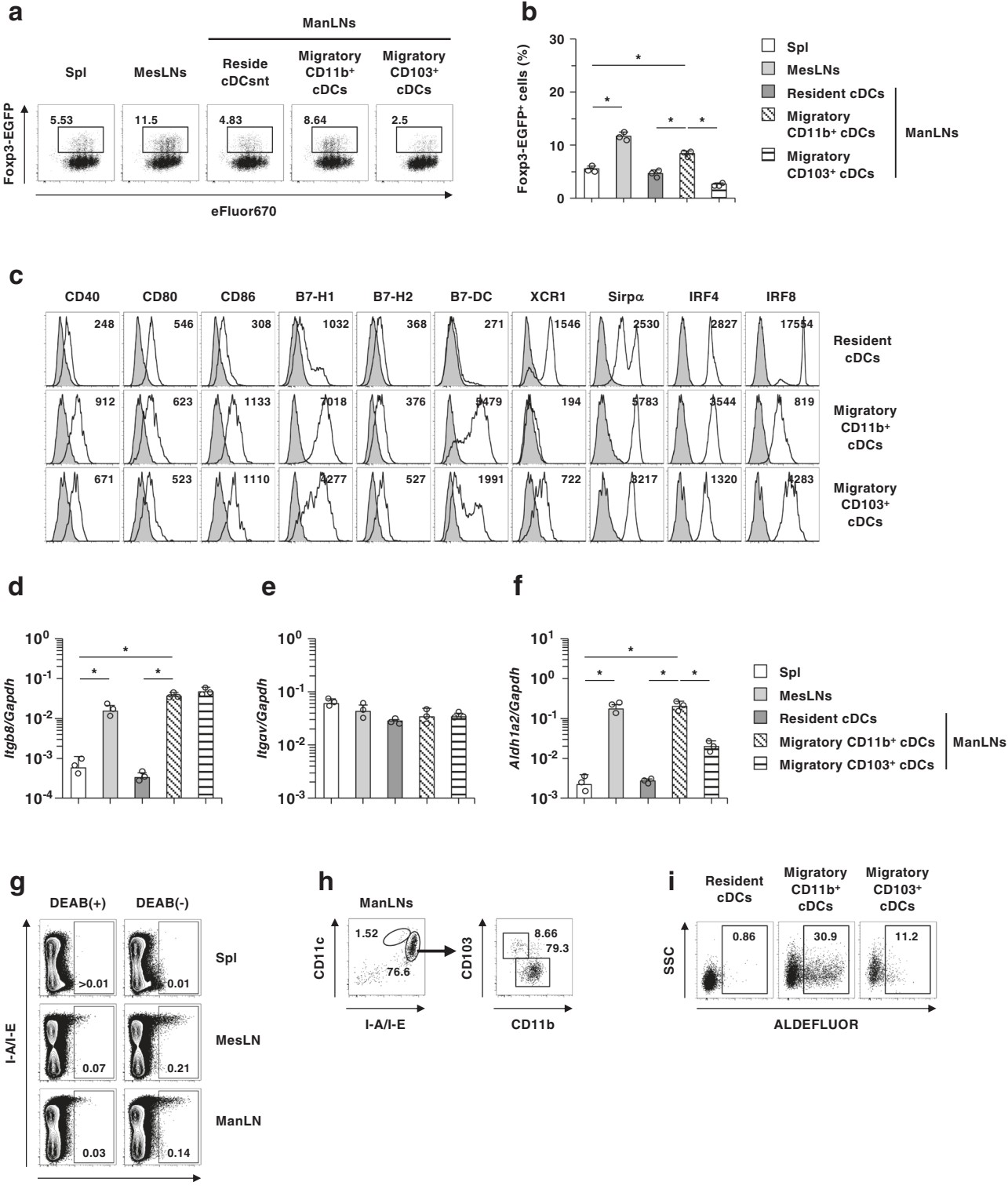

**Fig. 5 ManLNs migratory CD11b⁺ cDCs induces antigen-specific CD4⁺Foxp3⁺ pT_reg cells in vitro. a**, **b** eFluor™ 670-labeled KJ1–26⁺CD4⁺*Foxp3*^EGFP⁻ T cells were cultured with cDCs isolated from spleen, MesLNs, and ManLNs under pT_reg-polarized culture conditions for 5 days. Cell surface expression profile **a** and proportion **b** of KJ1–26⁺CD4⁺*Foxp3*^EGFP⁻ T cells. **c** Cell surface expression profile of ManLNs cDCs subset under steady-state condition, and numbers represent mean fluorescence intensity (MFI). **d–f** Transcriptional expressions of *Itgb8* **d**, *Itgav* **e**, and *aldh1a2* **f** in cDCs isolated from spleen, MesLNs, and ManLNs. **g–h** Leukocyes in spleen, MesLNs, and ManLNs obtained from WT mice were incubated with ALDEFLUOR in the presence or absence of diethylaminobenzaldehyde (DEAB) for detection of ALDH activity. **g** Cell surface expression profile of ALDEFLUOR⁺ cells in spleen, MesLNs, and ManLNs. **h**, **i** Cell surface expression profile of I-A/I-E^hiALDEFLUOR⁺ cells **h** and ALDEFLUOR⁺ cells in each cDCs subset in ManLNs **i**. Data are obtained from five individual samples in a single experiment. *$P < 0.05$ compared with among groups. All data are representative of at least three independent experiments.

effect of SLIT on type-I acute and chronic allergic disorders, where the tolerogenesis of migratory CD11b[+] cDCs are involved in the generation of antigen-specific CD4[+]Foxp3[+] T$_{reg}$ cells that inhibit the pathogenic T$_H$2-mediated allergic immune responses.

Although SLIT is widely introduced in the management of patients with allergic rhinitis and rhinoconjunctivitis as an alternative strategy to SCIT with rare incidence of adverse events, the adaptability of SLIT for the treatment of other type-I allergic disorders remains elusive. We showed that SLIT exhibited the protective effects on the development of allergic asthma as well as food allergy and systemic anaphylaxis in murine models, whereas the protective effect of SLIT was markedly abrogated under the elimination of cDCs. Therefore, SLIT may provide an advantageous means of antigen-specific intervention for T$_H$2-medaited acute and chronic type-I allergic disorders mediated through the function of cDCs. Further studies will be needed to determine the adaptation diseases of SLIT and to design its optimal treatment regimen, including the dose of allergen, the dosing schedule, and the treatment periods, in the clinical settings.

Relative to the better understanding of the immunological modification through SCIT, how SLIT controls allergic immune responses is less defined with some controversies. We showed that the sublingual antigenic application inhibited the generation of antigen-specific ST2[+] T$_H$2 cells in allergic inflammatory lesions and draining lymph nodes as well as the emergence of CXCR5[+]PD-1[+] T$_{FH}$ cells in secondary lymphoid tissues of asthmatic mice. Furthermore, the protected mice exhibited the reduced formation of the germinal center and the impaired generation of antigen-specific germinal center B cells in secondary lymphoid tissues as well as the diminished production of antigen-specific IgG$_1$ and IgE. Collectively, these results suggest that SLIT not only inhibits the development of the pathogenic T$_H$2 cells for the impaired formation of the inflammatory allergic pathogenesis but also suppressed the generation of T$_{FH}$ cells for the abortive generation of high-affinity Ab-producing plasma cells and memory B cells in germinal centers, and these immunological modifications leads to the amelioration of T$_H$2-medaited acute and chronic type-I allergic disorders.

Accumulating evidences indicate that there are some discrepancies between the apparent changes in immunological responses in peripheral blood and their manifestation of improved nasal and ocular symptoms in patients with allergic rhinitis and rhinoconjunctivitis received SLIT[34,35]. The reason why the effectiveness of SLIT for the treatment of these allergic disorders cannot coincide with the obvious changes of the immunological responses remains unclear, one possibility is that peripheral blood might not be suitable for monitoring the modifications of the immunological responses in patients treated with SLIT because the changes in immunological responses in peripheral blood does not necessarily reflect those in allergic inflammatory lesions and draining lymph nodes. Alternatively, the analytical methods might be technically insufficient to monitor the changes in immunological responses, such as sensitivity and the adaptation of the selected immunological parameters to detect the changes in humoral and cellular immune responses. On the other hand, we showed that the protective effect of SLIT on allergic disorders was correlated with the inhibition for the binding of the circulating basophils in peripheral blood with antigenic protein through the complexes of Fcε receptor Iα (FcεRIα)-IgE owing to the reduced production of antigen-specific IgE. Further technical advances might provide the optimal measurement of the reasonable immunological mediators and metabolites that are correlated with the protective effect of SLIT.

Although several types of the phagocytes with distinct tissue distributions have reportedly been existed in sublingual mucosa, the roles of APCs for capturing and transporting antigen to the draining ManLNs following sublingual antigenic application to initiate CD4[+] T-cell responses remain unclear. We showed that sublingual antigenic application exclusively induced antigen-specific proliferation of CD4[+] T cells in the draining ManLNs, but not in non-draining AurLNs and MesLNs, whereas it was severely diminished under the depletion of cDCs. Whereas migratory CD11b[+] cDCs predominantly retained antigen in ManLNs relative to other cDC subsets and leukocytes following sublingual antigenic application, the ablation of cDCs markedly reduced the accumulation of sublingual antigen in ManLNs. On the other hand, the depletion of macrophages had a minimal effect on antigen-specific proliferation of CD4[+] T cells in the draining ManLNs after sublingual antigenic administration. Taken together, these results suggest that CD11b[+] cDCs migrating from sublingual mucosa are critical APCs to present sublingual antigen for the priming of antigen-specific CD4[+] T cells in ManLNs.

Whereas the effectiveness of SCIT on allergic disorders has been considered to involve the modification of the deflection balance of T$_H$1/T$_H$2-responses, how SLIT improves T$_H$2-biased immune responses is not fully understood. We showed that sublingual antigenic application induced de novo generation of antigen-specific CD4[+]Foxp3[+] pT$_{reg}$ cells from CD4[+]Foxp3[−] T cells in ManLNs, but not in spleen and MesLNs, which was almost completely abolished under the elimination of cDCs. Furthermore, the accumulation of antigen-specific CD4[+]Foxp3[+] pT$_{reg}$ cells was observed in spleen following their emergence in ManLNs upon sublingual antigenic application, whereas treatment with FTY720 enhanced or reduced their accumulation in ManLNs or spleen, respectively. On the other hand, CD4[+]CD25[+] T$_{reg}$ cells, obtained from mice received SLIT inhibited the development of allergic pathogenesis, whereas their depletion abrogated the protective effect of SLIT. These observations support the hypothesis that SLIT initiates the generation of antigen-specific CD4[+]Foxp3[+] pT$_{reg}$ cells in ManLNs, and they subsequently disseminate into other peripheral lymphoid tissues via lymphatic vessels that inhibit the generation and function of pathogenic T$_H$2 and T$_{FH}$ cells for the amelioration of T$_H$2-medaited acute and chronic type-I allergic disorders. Collectively, our findings suggest that the mechanism responsible for the protective effect of SLIT on deleterious allergic diseases involves the mucosal tolerance that requires the existence of cDCs rather than the shift in the T$_H$1/T$_H$2 balance, and ManLNs serve as primary sites to generate antigen-specific CD4[+]Foxp3[+] pT$_{reg}$ cells for establishing mucosal tolerance, which is different process from the induction of oral tolerance that triggered in MesLNs upon exposure to dietary constituents in intestines.

Much attention has been paid to the functional characterization of intestinal cDC subsets compromised of CD103[+]CD11b[−] cDC1, CD103[+]CD11b[+] cDC2, and CD103[−]CD11b[+] cDC2 to clarify the mechanism responsible for the induction of oral tolerance[7–11], but far less is known about the role of ManLN DC subsets in the program for the establishment of mucosal tolerance. Similar to the functional differences of oral DC subsets[27], we showed that migratory CD11b[+] cDC2 displayed a higher capacity to drive TGF-β-mediated differentiation of antigen-specific CD4[+]Foxp3[+] pT$_{reg}$ cells than migratory CD103[+] cDC1 and resident cDCs in ManLNs as well as splenic cDCs, and that was similarly observed in MesLN cDCs. Furthermore, migratory CD11b[+] cDC2 imprint the characteristic features to favor the program for the generation of CD4[+]Foxp3[+] pT$_{reg}$ cells compared with other ManLN cDC subsets, including the preferential production of TGF-β and RA as well as the prominent expressions of B7-H1 and B7-DC. Taken together, these findings suggest the prerequisite role of CD11b[+] cDC2 migrating from sublingual

mucosa for establishing mucosal tolerance mediated through the de novo generation of antigen-specific CD4+Foxp3+ pT$_{reg}$ cells from CD4+Foxp3− T cells for the protective effect of SLIT on T$_H$2-mediated acute and chronic type-I allergic disorders. On the other hand, intestinal CD103+CD11b− cDC1 have shown to play a dominant role in de novo generation of CD4+Foxp3+ pT$_{reg}$ cells relative to CD103+CD11b+ cDC2 and CD103−CD11b+ cDC2 to establish oral tolerance possibly owing to their superior capacity to produce TGF-β and RA[7–11]. Although the precise reason for the distinct dominancy of tolerogenic function of cDC1 and cDC2 between intestinal tissues and ManLNs remains unclear, the observed discrepancy might be due to the influence of the different mucosal microenvironments on the development from their progenitors into each DC subset. Further study will be needed to examine this possibility.

In conclusion, we report that migratory CD11b+ cDC2 rather than other cDC subsets in ManLNs predominantly provide a milieu of T$_{reg}$-driven mucosal tolerance inhibiting allergic T$_H$2-responses that mediates the efficacy of SLIT on acute and chronic type-I allergic disorders. In future studies, a better understanding of the tolerogenesis of ManLN DC subsets and the identification of the eligible immunological parameters implicating the improvement of allergic symptoms may open new avenues for the incremental reliability of SLIT as prophylactic and therapeutic treatment for symptomatic patients with allergic rhinitis and rhinoconjunctivitis, and asymptomatic subjects sensitized to their allergens as well as clinical availabilities for other allergic diseases.

## Methods

**Mice**. The following 8- to 12-week-old female mice were used in this study. C57BL/6 and BALB/c mice were purchased from Japan Clea. B6.FVB-Tg$^{Itgax-DTR/EGFP}$ Lan/J (CD11c-DTR/EGFP) mice (referred to as cDC-ablated mice) were generated as described previously[28,29]. B6.CD45.1+OT-II OVA-specific TCR transgenic mice (B6. CD45.1+OT-II mice)[4], B6.CD45.1+Foxp3$^{EGFP}$OT-II OVA-specific TCR transgenic mice (B6.CD45.1+Foxp3$^{EGFP}$OT-II mice)[4], and Rag2$^{−/−}$Foxp3$^{EGFP}$DO11.10 OVA-specific TCR (KJ1–26 clonotype) transgenic BALB/c mice (Rag2$^{−/−}$Foxp3$^{EGFP}$ DO11.10 mice)[5,8] lacking KJ1–26+CD4+Foxp3$^{EGFP+}$ nT$_{reg}$ cells were generated as described previously. For the systemic ablation of cDCs, CD11c-DTR/EGFP mice were i.p. injected with DT (100 ng/mouse; Sigma-Aldrich) used as cDC-ablated mice. For the systemic ablation of CD4+Foxp3+ T$_{reg}$ cells, DEREG mice[33] were i.p. injected with DT (1 µg/mouse). In parallel experiments, CD11c-DTR/EGFP mice or DEREG mice were i.p. injected with phosphate-buffered saline (PBS) used as cDC- or CD4+Foxp3+ T$_{reg}$ cell-sufficient control mice. All mice were bred and maintained in specific pathogen-free conditions in the animal facility at the University of Miyazaki, and all experiments were performed in accordance with institutional guidelines of the Animal Experiment Committee and Gene Recombination Experiment Committee.

**Cell isolation**. Leukocytes were prepared from spleen, lymph nodes, lung, siLP, and colonic LP in mice as described below[5,29]. To prepare single-cell suspensions, spleen, lymph node, and lung were digested with 400 U/ml collagenase type III (Worthington Biochemical) at 37 °C for 20–40 min, and were ground between glass slides. Splenocytes were treated with red blood Cell lysis buffer (Sigma-Aldrich) before suspension. For the isolation of LP leukocytes, small or large intestines were opened longitudinally, washed to remove fecal content, and cut into small pieces. To remove epithelial cells, intestinal segments were treated in PBS containing 10% FCS, 20 mM HEPES, 100 U/ml penicillin, 100 µg/ml streptomycin, 1 mM sodium pyruvate, 20 mM ethylenediaminetetraacetic acid and 10 µg/ml polymyxin B (Calbiochem) with continuous stirring at 37 °C for 20 min in a water bath. After wash with PBS, remaining tissues ware incubated with 400 U/ml collagenase type III, 250 mU/ml dispase (Life Technologies), and 100 µg/ml DNase I (Roche Diagnostics) for 20–30 min at 37 °C in a water bath. The cell suspension was prepared by forcing through a 100-µm cell strainer, washed with PBS, resuspended in 10 ml of 30% Percoll (GE Healthcare) and overlaid on 2 ml of 70% percoll in a 15-ml tube. Percoll gradient separation was performed by centrifugation at 780 g for 20 min at room temperature. The LP leukocytes were collected at the interface of the Percoll gradient and washed with RPMI1640/10% FCS, and used immediately for experiments. Murine peripheral blood mononuclear cells (PBMCs) were collected from blood by the use of Lympholyte-M (Sedarlane). For the isolation of murine BAL leukocytes, the trachea was cannulated and lavaged two times with 1 ml sterile cold PBS. For the isolation of oral macrophages, oral tissues were collected and treated with collagenase type III (400 U/ml; Warthington), dispase (250 mU/ml; Life Technologies) and DNase I (100 ng/ml; Sigma-Aldrich) at 37 °C for 45 min. Murine CD11c+ DCs were purified by AutoMACS with mouse CD11c

(N418) Microbeads (Miltenyi Biotec). Subsequently, ManLNs CD11c+ DCs were sorted into migratory I-A/I-E$^{hi}$CD11c$^{med}$CD11b+ cDCs, I-A/I-E$^{hi}$CD11c$^{med}$ CD103+ cDCs and resident I-A/I-E$^{med}$CD11c$^{hi}$ cDCs with high purity (each > 99%) using a FACSAriaII cell sorter (BD Biosciences) with fluorescein-conjugated mAbs (BD Biosciences). Murine CD4+ T cells were purified from splenocytes of WT mice, B6.CD45.1+OT-II mice (CD45.1+Vα2+CD4+ T cells), B6. CD45.1+Foxp3$^{EGFP}$OT-II mice (CD45.1+Foxp3$^{EGFP}$Vα2+CD4+ T cells), and Rag2$^{−/−}$Foxp3$^{EGFP}$DO11.10 mice (KJ1–26+CD4+Foxp3$^{EGFP−}$ T cells) with mouse CD4 T lymphocyte Enrichment Set-DM (BD Biosciences). In some experiments, murine CD4+ T cells were sorted into CD4+CD25+ T cells and CD4+CD25− T cells with high purity (each >99%) by a FACSAriaII cell sorter with fluorescein-conjugated mAbs (BD Biosciences). Similarly, CD45.1+Vα2+CD4+Foxp3$^{EGFP}$ T cells (CD45.1+OT-II CD4+Foxp3$^{EGFP−}$ T cells) were purified from CD45.1+Foxp3$^{EGFP}$Vα2+CD4+ T cells with >99% purity by FACSAriaII cell sorter, respectively.

**Flow cytometry**. Murine cells were stained with fluorescein-conjugated mAbs listed in Supplementary Table 1. For the intracellular expression of cytokines[5], murine cells were incubated for 4 h with phorbol 12-myristate 13-acetate (PMA, 50 ng/ml; Sigma-Aldrich) and ionomycin (IoM, 500 ng/ml; Sigma-Aldrich) plus GolgiPlug (BD Biosciences) during the final 2 h. Subsequently, the cells were resuspended in fixation-permeabilization solution (eBiosciences) and intracellular cytokine staining was carried out according to the manufacturer's directions. In some experiments, murine CD4+ T cells ($5 \times 10^5$) were cultured with splenic CD11c+ DCs ($5 \times 10^4$) in the presence of OVA peptide (ISQAVHAAHAEI-NEAGR, 1 µM) for 5 days before the detection intracellular expression of cytokines. To detect ALDH activity[5], murine cells were stained with an ALDEFLUOR staining kit (StemCell Technologies) with or without the ALDH inhibitor diethy-laminobenzaldehyde (DEAB, 100 µM; Sigma-Aldrich) according to the manufacturer's protocol. Fluorescence staining was analyzed with a FACSVerse flow cytometer (BD Biosciences) and FlowJo software (Tree star).

**Quantitative reverse transcription polymerase chain reaction (RT-PCR)**. Total RNA from murine cells was extracted by using RNeasy plus micro kit (Qiagen) and the first-strand complementary DNA (cDNA) was synthesized from 100 ng of total RNA with oligo(dT)$_{20}$ primer using the PrimeScript RT reagent kit (Takara) according to the manufacturer's instructions. Transcriptional expression levels were analyzed as described previously[5] by using SYBR® Premix Ex Taq II on Thermal Cycler Dice (Takara) with specific primer pairs for Itgb8 (5′-acagcatcg-catggaccaa-3′ and 5′-aagcaacccgatcaagaatgtg-3′), Itgav (5′-ctgtggagataagaggagtttca-3′ and 5′-cccaacgtcttcttcagtct-3′), Aldh1a2 (5′-tgggtgagtttggcttacgg-3′ and 5′-agaaacgtggcagtcttggc-3′), and Gapdh (5′-aaattcaacggcacagtcaag-3′ and 5′-tggtggtgaagacaccagtag-3′) after normalization for Gapdh expression.

**Measurement of serum OVA-specific Ab**. Serum total IgG$_1$, OVA-specific IgG$_1$, total IgE, and OVA-specific IgE were assayed by using IgG$_1$ Mouse Uncoated (enzyme-linked immunosorbent assay) ELISA Kit (eBioscience), Mouse Anti-OVA-IgG$_1$ ELISA KIT (Shibayagi), Mouse IgE ELISA MAX kit, and a Mouse OVA-Specific IgE ELISA Kit (both from BioLegend) according to the manufacturer's instructions.

**Measurement of serum IL-5**. Serum IL-5 was measured by Flow cytometry with LEGENDplex™ Mouse Th Cytokine panel (BioLegend) according to the manufacturer's instructions.

**Sublingual administration with OVA protein**. For SLIT, mice were administered sublingually with or without 500 µg of OVA protein (A7642–1VL, Sigma-Aldrich) for 3 consecutive days a week during 3 weeks or 50 µg of Alexa Fluor 647-conjugated OVA dissolved in 10 µl PBS containing 3% carboxymethyl cellulose under sevoflurane anesthesia. In some experiments, CD11c-DTR/EGFP mice were i.p. injected with DT as described above on 1 day before each consecutive sub-lingual administration. Similarly, DEREG mice were i.p. injected with DT as described above on days 21 and 22 after the initial sublingual administration. Alternatively, WT mice were sublingually administrated with CL-A (100 µg/mouse; FormuMax Scientific) under sevoflurane anesthesia on 1 day before the sublingual administration for the depletion of macrophages.

**Airway inflammation to OVA**. On days 7 and 14 after SLIT, mice were i.p. immunized with 100 µg of OVA protein mixed in 4 mg of alum (Imject Alum; Thermo Scientific). Subsequently, mice were intranasally challenged with 100 µg OVA protein in 20 µl of PBS on days 10, 11, and 12 after the last immunization. At 13 days after the last immunization, spleen, serum, BAL, lung, and mediastinal lymph nodes were obtained from the mice[5]. Alternatively, spleen was obtained at 7 days after final immunization for immunohistochemical analysis. For measurement of airway responsiveness[5], airway function was analyzed for changes in R$_{rs}$ in response to increasing doses of inhaled methacholine (3.125, 6.25, 12.5, 25, and 50 mg/ml; Sigma-Aldrich) by using an invasive mechanical ventilator flexiVent (SCIREQ Scientific Respiratory Equipment).

**Food allergy to OVA.** On days 7 and 21 after SLIT, mice were i.p. immunized with 100 μg of OVA protein mixed in 4 mg of alum. From 14 days after the last immunization, mice were i.g. challenged with 50 mg OVA protein in saline every 2 days at a total of eight times. Before each i.g. challenge, mice were deprived of food for 2–3 h. Diarrhea (score: 0–4) was scored by visually monitoring mice for 60 min after i.g. challenge[36]. At 2 days after the last i.g. challenge, SI and colon were obtained from the mice.

**Systemic anaphylaxis to OVA.** On days 7 and 14 after SLIT, mice were i.p. immunized with 100 μg of OVA protein mixed in 4 mg of alum. At 10 days after the last immunization, mice were challenged by i.p. injection with 100 μg of OVA protein, and rectal temperature was measured by a digital thermometer (TD-300; Shibaura Electronics) every 5 min from 5 min after challenge for 60 min[5]. Alternatively, whole-body temperature was imaged 35 min after challenge by a thermography device (Handy Thermo TSV-200 ME, Nippon Avionics)[5]. In some experiments, PBMCs obtained from mice at 1 day before the challenge were cultured with FITC-conjugated OVA (2 μg/ml; Sigma-Aldrich) for 30 min. Subsequently, the specific binding of FITC-conjugated OVA to $CD3\varepsilon^- B220^- CD11b^- Fc\varepsilon RI\alpha^+ CD49b^+$ basophils was analyzed by flow cytometry. Correlation between MFI for the binding of FITC-conjugated OVA and decline in body temperature (Δ°C) was calculated to determine $R^2$. Serum were obtained at 1 day before the challenge to measure OVA-specific Ab or 90 s after the challenge to detect histamine.

**Adoptive transfer.** For antigen-specific priming of murine CD4[+] T cells in vivo[5,8], $CD45.1^+$OT-II CD4[+] T cells were labeled with eFluor™ 670 (Thermo Fisher Scientific; 2.5 μM) at 37 °C for 10 min, and washed twice with cold PBS. Subsequently, eFluor™ 670-labeled $CD45.1^+$OT-II CD4[+] T cells ($5 \times 10^6$/mouse) were intravenously (i.v.) injected into mice 24 h before sublingual application with OVA protein. After 3 days, the gated $CD45.1^+$OT-II CD4[+] T cells in spleen, AurLNs and ManLNs were analyzed for eFluor™ 670 dilution to detect the dividing cells by flow cytometry. In some experiments, mice were i.p. injected with CL-A (100 μg/mouse) on 1 day before adoptive transfer of $CD45.1^+$OT-II CD4[+] T cells. For the differentiation of murine $Foxp3^{EGFP+}$ pT$_{reg}$ cells in vivo, mice were i.v. injected with $CD45.1^+$OT-II $CD4^+Foxp3^{EGFP-}$ T cells ($5 \times 10^6$/mouse), and then sublingually administered with or without OVA protein for three consecutive days as described above. Alternatively, mice received daily i.p. injection of FTY720 (1 mg kg$^{-1}$ per BW; Cayman chemical) from 1 day before SLIT to day 6 after the start of the SLIT treatment. On day 5 or 7, the expression of $Foxp3^{EGFP}$ among gated $CD45.1^+$OT-II CD4[+] T cells was analyzed by flow cytometry. In another experiment, $CD4^+CD25^+$ or $CD4^+CD25^-$ T cells ($2.5 \times 10^6$/mouse) obtained from SLIT-treated mice or naive mice were i.v. injected into mice before immunization with OVA protein plus alum adjuvant.

**In vitro CD4[+] T-cell differentiation assay.** For the differentiation of murine $Foxp3^{EGFP+}$ pT$_{reg}$ cells[5,8], KJ1–26$^+$CD4$^+Foxp3^{EGFP-}$ T cells ($5 \times 10^4$) were cultured with splenic cDCs ($5 \times 10^3$) in the presence or absence of recombinant human TGF-β1 (20 ng/ml) in combination with OVAp (1 μM), anti-IFN-γ mAb (10 μg/ml; R4–6A2, BD Biosciences), anti-IL-4 mAb (10 μg/ml; 11B11, BD Biosciences), and recombinant mouse IL-2 (0.2 ng/ml; Wako Pure Chemicals) for 5 days in 96-well round-bottomed plates (BD Biosciences). Alternatively, KJ1–26$^+$CD4$^+Foxp3^{EGFP-}$ T cells were labeled with eFluor™ 670 as described above prior to pT$_{reg}$-polarized culture conditions. In some experiments, control IgG, anti-B7-H1 (10 μg/ml; M1H5), B7-H2 (10 μg/ml; HK5.3) and B7-DC (10 μg/ml; TY25) (eBioscience) were added to the culture. Analysis of the expression of $Foxp3^{EGFP}$ among CD4[+] T cells was performed by flow cytometry as described above.

**Analysis of T$_{reg}$-cell function.** For preparation of murine KJ1–26$^+$CD4$^+$ST2$^+$CD44$^+$ T cells, $Rag2^{-/-}Foxp3^{EGFP}$DO11.10 mice received twice i.p. immunization with 100 μg of OVA protein mixed in 4 mg of alum at 7-day interval followed by intranasal challenge with 100 μg OVA protein in 20 μl of PBS on days 10, 11, and 12 after the last immunization. At 13 days after the last immunization, spleens were obtained from the mice as described above. Subsequently, KJ1–26$^+$CD4$^+$ST2$^+$CD44$^+$ T cells were purified from KJ1–26$^+$CD4$^+Foxp3^{EGFP-}$ T cells with >99% purity by FACSAriaII cell sorter. For analysis of T$_{reg}$-cell function, eFluor™ 670-labeled KJ1–26$^+$CD4$^+$ST2$^+$CD44$^+$ T cells ($5 \times 10^4$) were cultured with splenic cDCs ($5 \times 10^3$) in the presence or absence of $CD4^+CD25^+$ or $CD4^+CD25^-$ T cells ($2.5 \times 10^4$) obtained from SLIT-treated mice or naive mice as described above. After 3 days, the gated KJ1–26$^+$CD4[+] T cells were analyzed for eFluor™ 670 dilution to detect the dividing cells by flow cytometry.

**Histopathologic assessment.** Tissues from the lung and SI were fixed with 4% paraformaldehyde in PBS and embedded in paraffin. The lung tissue sections (5 μm) were stained with Hematoxylin and eosin and Periodic acid–Schiff[5]. Intestinal mast cells were stained with chloro-acetate esterase staining kit (Sigma-Aldrich; 91C-1KT) according to the manufacturer's guidance[37]. The stained slides were examined with a bright-field microscopy (BX53; Olympus).

**Immunohistochemical analysis.** Spleen and ManLNs were embedded in OCT compound (Sakura Finetech) and frozen in liquid $N_2$. Frozen section (5 μm) was fixed with cold acetone, and blocked with PBS containing 5% of normal rat serum and Avidin/Biotin Blocking Kit (Vector laboratories). Subsequently, slide was stained with Alexa Fluor 488-conjugated anti-B220/CD45R mAb, biotin-conjugated PNA (MBL, Nagoya, Japan), followed by Alexa Fluor 546-conjugated Streptavidin (Invitrogen) and Alexa Fluor 647-conjugated anti-CD3ε mAb, and mounted with ProLong Diamond (Thermo Fisher Scientific). The stained slides were analyzed with BZ-X710 fluorescence microscope (KEYENCE, Osaka, Japan).

**Statistics and reproducibility.** Data are expressed as the mean ± s.d from three to ten individual samples in a single experiment, and we performed at least three independent experiments. The statistical significance of the differences between the values obtained was evaluated by two-sided paired student $t$ test. A $P$ value of < 0.05 was considered significant.

**Reporting summary.** Further information on research design is available in the Nature Research Reporting Summary linked to this article.

## Data availability

The data sets that support the finding of this study are available from the corresponding author on reasonable request.

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

## Acknowledgements

We thank all members of the animal facility at University of Miyazaki; Yumiko Sato and Ikumi Goto for secretarial assistance and Yukari Kawagoe for technical help in cell sorting. This work was supported by a Grant-in-Aid for Scientific Research (B) (K.S.; 18H02670), for challenging Exploratory Research (K.S.; 16K15291), and for Young Scientists (B) (T.U.; 17K15027, T. Fukaya.; 17K15732, and H.T.; 18K15194) from the Ministry of Education, Science and Culture of Japan, the Project for Cancer Research And Therapeutic Evolution (P-CREATE) from Japan Agency for Medical Research and development (AMED) (K.S.; 16cm0106307h0001, 17cm0106307h0002, 18cm0106307h0003, and 19cm0106307h0004), the Uehara Memorial Foundation (H.T.), Takeda Science Foundation (T.U. and T. Fukaya.), the Naito Foundation (K.S.), Bristol-Myers Squibb Foundation Grants (K.S.), GSK Japan Research Grant 2016 (T. Fukaya.), GSK Japan Research Grant 2017 (H.T.), GSK Japan Research Grant 2018 (T.U.), Daiichi Sankyo Foundation of Life Science (K.S.), Nipponham Foundation for the Future of Food (H.T.), and the Shin-Nihon Foundation of Advanced Medical Research (T.U.).

## Author contributions

K.S. designed all experiments, analyzed data, and wrote the manuscript, N.M., H.T., T.U., T. Fukaya, J.N., T. Fukui, Y.N. and N.C. performed experiments, and T.S., Y.H., T.N., and T.T. provided reagents and information.

## Competing interests

The authors declare no competing interests.
