## [Peer Review File · Communications Biology]

Reviewers' Comments:

Reviewer #1:

Remarks to the Author:

Summary

Prophylactic treatment with sublingual immunotherapy (SLIT) for allergic mouse models of asthma, food allergy, or anaphylaxis. The authors treated mice with SLIT before development of allergic disorders, and showed that SLIT is effective for the protection against the development of allergic disorders. The authors demonstrated the immunological mechanism of the prophylactic SLIT. The authors showed that conventional DCs (cDCs) and CD4⁺Foxp3⁺ Treg cells were required for the SLIT, and showed sublingual Ag administration induced Ag-specific CD4⁺Foxp3⁺ Tregs, and cDCs were required for the CD4⁺Foxp3⁺ Treg induction in the submandibular lymph nodes. The author also showed that CD11b⁺ cDC subset was superior to other cDC subset for the generation of CD4⁺Foxp3⁺ Tregs.

Overall impressions

The authors showed SLIT had a potential to protect the development of Ag-specific allergic disorders by using mouse model. It is interesting to examine the immunological mechanisms in the experimental prophylactic SLIT. Especially, the author clearly showed the induction of Ag-specific Tregs after sublingual Ag application, and the roles and characteristics of cDC subset in the induction of Tregs. These results may be a reference to understand the mechanism of conventional therapeutic administration of SLIT as well.

Specific comments

1) I think it is more appropriate to add the phrase "prophylactic" or "protective" before "sublingual immunotherapy" in the title.

2) What mouse strain were used in Fig1 e-h, j-l? Balbc/WT?

3) The authors described the aim is to clarify the contribution of cDCs to the effectiveness of SLIT protecting against Th2-mediated allergic air way responses in P6.L3-4 and concluded the absence of cDCs impairs protective effect of SLIT on the development of these Th2-mediated allergic pathogenesis in P6L16-17 and P7L8-9 by two results that the number of eosinophils in BALF cells and systemic anaphylaxis were not recovered with SLIT in CD11c-DTR/EGFR Tg mice with DT. However, I think the contributions of cDCs were not fully shown to the effectiveness of SLIT protecting against Th2-mediated allergic responses in Fig 1. What were the results regarding Th2 responses such as IgG1, IgE, airway function, or Ag-specific Th2 cell responses in CD11c-DTR/EGFR Tg mice?

4) The author showed the generation of ST2⁺CD4⁺pathogenic T cells were suppressed in the asthmatic mice and mice with food allergy with prophylactic SLIT in Fig2 a-c and sFig2 d-f. And in conclusion, the author described migratory CD11b⁺cDCs are involved in the generation of Ag-specific CD4⁺Foxp3⁺ Tregs that inhibit the pathogenic Th2-mediated allergic immune response P13L4-6. However, the contribution of pathogenic ST2⁺ T cells on the pathogenesis of allergic disorders in the model mouse, the relationship between pathogenic T cells and Ag-specific CD4⁺Foxp3⁺ Tregs or cDCs, or the molecular and immunological mechanisms were not examined in this study. To make the conclusion, I think the author should show some mechanism.

5) The author clearly showed that sublingually administrated OVA-Ag was captured and retained in migratory cDCs in ManLNs in Fig3d. The authors showed OVA-specific T cells showed division in ManLNs and OVA-specific Foxp3⁺CD4⁺ T cells were induced in MnLNs mediated by cDC after sublingual Ag administration in Fig4. Do the induced Foxp3⁺ CD4⁺ T cells have suppressive function? specific Th2 cell response was suppressed in the mouse model or ex vivo?

6) The author used CD4⁺CD25⁺Treg cells from mice received Ag application for transfer

experiment in Fig 4i and sFig6d. Was there difference in the suppressive effect on systemic anaphylaxis by transfer of CD4+CD25+Treg cells from mice between with and without Ag application?

Reviewer #2:

Remarks to the Author:

The manuscript by Miyanaga, et. al. reports that migratory CD11b+ cDCs in ManLNs provide a milieu of Treg-driven mucosal tolerance inhibiting allergic Th2-responses that mediates the efficacy of SLIT on acute and chronic type I allergic disorders. The research topic is very important. The experiments conducted are detailed and thorough and the mechanistic findings are novel and intriguing. Despite the enthusiasm, a few concerns need to be addressed by the authors, before the manuscript can be considered for publication.

1. In Fig. 1, the authors need to detect the Th2 cytokine expression (such as IL4 or IL13) by ELISA to see if deficiency of cDCs really attenuates the protective effect of SLIT on OVA induced Th2-mediated allergic airway inflammation.
2. In Fig. 2i, the authors need to include the scale bar for the IFA figures.
3. In the figure legend of Fig. 1j, there are mistakes for the methacholine doses (j; 3.125-50 mg/ml). The authors need to revise it.
4. In the figure legend of Supplementary Fig.2, the authors need to change the capital A-F into small.

Response to Reviewers

Response to the reviewer #1

We carefully revised the manuscript according to the reviewers' comments and suggestions. The revised portions are in red font. Detailed responses are indicated below.

Summary

Prophylactic treatment with sublingual immunotherapy (SLIT) for allergic mouse models of asthma, food allergy, or anaphylaxis. The authors treated mice with SLIT before development of allergic disorders, and showed that SLIT is effective for the protection against the development of allergic disorders. The authors demonstrated the immunological mechanism of the prophylactic SLIT. The authors showed that conventional DCs (cDCs) and CD4+Foxp3+ Treg cells were required for the SLIT, and showed sublingual Ag administration induced Ag-specific CD4+Foxp3+ Tregs, and cDCs were required for the CD4+Foxp3 Treg induction in the submandibular lymph nodes. The author also showed that CD11b+ cDC subset was superior to other cDC subset for the generation of CD4+Foxp3+ Tregs.

Overall impressions

The authors showed SLIT had a potential to protect the development of Ag-specific allergic disorders by using mouse model. It is interesting to examine the immunological mechanisms in the experimental prophylactic SLIT. Especially, the author clearly showed the induction of Ag-specific Tregs after sublingual Ag application, and the roles and characteristics of cDC subset in the induction of Tregs. These results may be a reference to understand the mechanism of conventional therapeutic administration of SLIT as well.

Specific comments

1) I think it is more appropriate to add the phrase “prophylactic” or “protective” before “sublingual immunotherapy” in the title.

-As suggested, we added “protective” before “sublingual immunotherapy” in the title in the revision.

2) What mouse strain were used in Fig1 e-h, j-l? Balbc/WT?

-We used Balb/c WT mice. As suggested, we added “in Balb/c WT mice” in the Figure 1 legend in the revision.

3) The authors described the aim is to clarify the contribution of cDCs to the effectiveness of SLIT protecting against Th2-mediated allergic air way responses in P6.L3-4 and concluded the absence of cDCs impairs protective effect of SLIT on the development of these Th2-mediated allergic pathogenesis in P6L16-17 and P7L8-9 by two results that the number of eosinophils in BALF cells and systemic anaphylaxis were not recovered with SLIT in CD11c-DTR/EGFR Tg mice with DT. However, I think the contributions of cDCs were not fully shown to the effectiveness of SLIT protecting against Th2-mediated allergic responses in Fig 1. What were the results regarding Th2 responses such as IgG1, IgE, airway function, or Ag-specific Th2 cell responses in CD11c-DTR/EGFR Tg mice?

-As suggested, we showed the data regarding Th2 responses such as IgG1, IgE, airway function, or Ag-specific Th2 cell responses in CD11c-DTR/EGFR Tg mice in Supplementary Fig. 2 in the revision.

4) The author showed the generation of ST2⁺CD4⁺pathogenic T cells were suppressed in the asthmatic mice and mice with food allergy with prophylactic SLIT in Fig2 a-c and sFig2 d-f. And in conclusion, the author described migratory CD11b⁺cDCs are involved in the generation of Ag-specific CD4⁺Foxp3⁺ Tregs that inhibit the pathogenic Th2-mediated allergic immune response P13L4-6. However, the contribution of pathogenic ST2⁺ T cells on the pathogenesis of allergic disorders in the model mouse, the relationship between pathogenic T cells and Ag-specific CD4⁺Foxp3⁺ Tregs or cDCs, or the molecular and immunological mechanisms were not examined in this study. To make the conclusion, I think the author should show some mechanism.

-We appreciate your comments. Previous studies have shown that the contribution of pathogenic ST2⁺ T cells on the pathogenesis of allergic disorders in the model mouse (e.g., ref. 30). Regarding immunological mechanisms for the protective effect of SLIT on the development allergic disorders, we showed that 1) the absence of cDCs impaired the protective effect of SLIT against the generation of pathogenic ST2⁺CD4⁺ T cells (response to the reviewer's comment #3), Ag-specific B-cell responses (response to the reviewer's comment #3), and T_H2-mediated allergic pathogenesis, 2) the ablation of CD4⁺Foxp3⁺ T_{reg} cells suppressed the protective effect of SLIT on the development of T_H2-mediated allergic pathogenesis, 3) the deficiency of cDCs inhibited the induction of Ag-specific CD4⁺Foxp3⁺ T_{reg} cells in ManLNs where are the primary sites in their generation upon sublingual antigenic application, and 4) CD4⁺Foxp3⁺ T_{reg} cells generated following sublingual antigenic application suppressed the activation of Ag-specific ST2⁺CD4⁺ T cells (response to the reviewer's comment #5). Therefore, we described the sentence in P13L4-6 in original manuscript. Regarding the molecular mechanisms, we focused on the characteristic features of ManLN cDC subsets for the

Response to Reviewers

induction of CD4⁺Foxp3⁺ T_{reg} cell. As described in original manuscript, migratory CD11b⁺ cDC2 imprint the characteristic features to favor the program for the generation of CD4⁺Foxp3⁺ pT_{reg} cells (ref. 7-11) compared with other cDC subsets in ManLNs, including the preferential production of TGF-β and RA as well as the prominent expressions of B7-H1 and B7-DC. Because we might not show the sufficient mechanism requested by the reviewer to prove the direct evidences for the relationship between pathogenic T cells and Ag-specific CD4⁺Foxp3⁺ Tregs or cDCs, we changed the “our findings propose” from “we revealed” in the section of Discussion in the revision.

5) The author clearly showed that sublingually administrated OVA-Ag was captured and retained in migratory cDCs in ManLNs in Fig3d. The authors showed OVA-specific T cells showed division in ManLNs and OVA-specific Foxp3⁺CD4⁺ T cells were induced in MnLNs mediated by cDC after sublingual Ag administration in Fig4. Do the induced Foxp3⁺ CD4⁺ T cells have suppressive function? specific Th2 cell response was suppressed in the mouse model or ex vivo?

-As suggested, we showed that CD4⁺CD25⁺ T_{reg} cells, but not CD4⁺CD25⁻ T cells obtained from mice received sublingual antigenic application exhibited a more inhibitory effect on Ag-specific proliferation of KJ1-26⁺CD4⁺ST2⁺CD44⁺ T cells than CD4⁺CD25⁺ T_{reg} cells from naïve mice in Supplementary Fig. 7e-g in the revision.

6) The author used CD4⁺CD25⁺Treg cells from mice received Ag application for transfer experiment in Fig 4i and sFig6d. Was there difference in the suppressive effect on systemic anaphylaxis by transfer of CD4⁺CD25⁺Treg cells from mice between with and without Ag application?

Response to Reviewers

-Regarding Fig. 4i, the adoptive transfer with CD4⁺CD25⁺ T_{reg} cells obtained from mice received sublingual antigenic application exhibited a more potent protection against the development of systemic anaphylaxis than CD4⁺CD25⁺ T_{reg} cells from naive mice. We added the statistics symbol (†) between CD25⁺ from SLIT mice and CD25⁺ from naive mice in Fig. 4i in the revision.

Response to the reviewer #2

We carefully revised the manuscript according to the reviewers' comments and suggestions. The revised portions are in red font. Detailed responses are indicated below.

The manuscript by Miyanaga, et. al. reports that migratory CD11b⁺ cDCs in ManLNs provide a milieu of Treg-driven mucosal tolerance inhibiting allergic Th2-responses that mediates the efficacy of SLIT on acute and chronic type I allergic disorders. The research topic is very important. The experiments conducted are detailed and thorough and the mechanistic findings are novel and intriguing. Despite the enthusiasm, a few concerns need to be addressed by the authors, before the manuscript can be considered for publication.

1. In Fig. 1, the authors need to detect the Th2 cytokine expression (such as IL4 or IL13) by ELISA to see if deficiency of cDCs really attenuates the protective effect of SLIT on OVA induced Th2-mediated allergic airway inflammation.

-Possibly due to the B6-background, we could not detect serum production of IL-4 and IL-13 in CD11c-DTR/EGFP mice that had received immunization with OVA protein plus alum adjuvant. On the other hand, the sublingual administration with OVA protein

Response to Reviewers

before immunization reduced serum production of IL-5 in CD11c-DTR/EGFP mice that had been treated with PBS, whereas the elimination of cDCs abrogated the suppressive effect of SLIT on serum production of IL-5 in the immunized CD11c-DTR/EGFP mice that had been treated with DT. As suggested, we showed the data regarding serum production of IL-5 in Supplementary Fig. 2e in the revision.

2. In Fig. 2i, the authors need to include the scale bar for the IFA figures.

-As suggested, we added the scale bar in Fig. 2i in the revision.

3. In the figure legend of Fig. 1j, there are mistakes for the methacholine doses (j; 3.125-50 mg/ml). The authors need to revise it.

-Although the reviewer #2 pointed out the methacholine doses, they are the correct values according to our published reports (Fig. 6A in J. Allergy Clin. Immunol. 2018;141:2156-2167)

4. In the figure legend of Supplementary Fig.2, the authors need to change the capital A-F into small.

-As pointed out, we corrected them in Supplementary Fig. 3 in the revision.

Reviewers' Comments:

Reviewer #1:

Remarks to the Author:

I think the authors properly revised the manuscript.

Reviewer #2:

Remarks to the Author:

The points raised in the previous round of review have been satisfactorily addressed.